# Exact Results for the Distribution of Randomly Weighted Sums

**Thomas Hitchen and Saralees Nadarajah ***

Department of Mathematics, University of Manchester, Manchester M13 9PL, UK;
thomas.hitchen@manchester.ac.uk
* Correspondence: mbbsssn2@manchester.ac.uk

**Abstract:** Dependent random variables play a crucial role in various fields, from finance and statistics to engineering and environmental sciences. Often, interest lies in understanding the aggregate sum of a collection of dependent variables with random weights. In this paper, we provide a comprehensive study of the distribution of the aggregate sum with random weights. Expressions derived include those for the cumulative distribution function, probability density function, conditional expectation, moment generating function, characteristic function, cumulant generating function, moments, variance, skewness, kurtosis, cumulants, value at risk and the expected shortfall. Real data applications are discussed.

**Keywords:** conditional expectation; copulas; cumulative distribution function; probability density function

**MSC:** 62E99

## 1. Introduction

There have been many papers studying the distribution of randomly weighted sums. Papers published since 2020 have studied: inequalities for sums of randomly weighted random variables [1–3]; randomly weighted sums of conditionally dependent and dominated varying-tailed increments [4]; second-order tail behavior of randomly weighted heavy-tailed sums [5]; complete and complete moment convergence for randomly weighted sums [6]; approximations for the tail behavior of bidimensional randomly weighted sums [7,8]; complete convergence for randomly weighted sums of random variables satisfying some moment inequalities [9]; complete convergence and complete moment convergence for maximal randomly weighted sums [10]; asymptotic distributions of randomly weighted sums [11]; complete moment convergence for randomly weighted sums of extended negatively dependent sequences [12]; complete convergence for randomly weighted sums [13,14]; complete $f$-moment convergence for randomly weighted sums [15]; tail asymptotics of randomly weighted sums of dependent strong subexponential random variables [16]; sums of two dependent randomly weighted random variables [17]; tail behavior of randomly weighted sums of dependent subexponential random variables [18]; randomly weighted sums for multivariate Dirichlet distributions [19]; complete convergence and complete integral convergence for randomly weighted sums [20]; complete moment convergence for randomly weighted sums of negatively superadditive-dependent random variables [21]; complete moment convergence for the maximum of randomly weighted sums [22]; the Baum–Katz theorem for randomly weighted sums [23]; asymptotics for the joint tail probability of bidimensional randomly weighted sums [24]; complete moment convergence for randomly weighted sums [25]. Other highly cited papers studying the distribution of randomly weighted sums include [26–30].

But, none of these papers study the exact distribution of randomly weighted sums without considering particular distributions. In this paper, we provide a comprehensive study of the exact distribution of $Z = W_1 X_1 + \cdots + W_n X_n$, where $X_1, \ldots, X_n$ are dependent

with a copula-induced dependency and the weights $W_1, \ldots, W_n$ are random and subject to the condition that they sum to 1. We consider four situations: (i) $(X_1, \ldots, X_n)$ are dependent and $(W_1, \ldots, W_{n-1})$ are independent; (ii) $(X_1, \ldots, X_N)$ are dependent conditioned on $N$, $(W_1, \ldots, W_{N-1})$ are independent conditioned on $N$ and $N$ itself is a random variable; (iii) $(X_1, \ldots, X_n)$ are dependent and $(W_1, \ldots, W_{n-1})$ are dependent too; (iv) $(X_1, \ldots, X_N)$ are dependent conditioned on $N$, $(W_1, \ldots, W_{N-1})$ are dependent too conditioned on $N$ and $N$ itself is a random variable. For each of these situations, we derive formulae for the cumulative distribution function, probability density function, conditional expectation, moment generating function, characteristic function, cumulant generating function, moments, variance, skewness, kurtosis, cumulants, value at risk and the expected shortfall of $Z$.

Section 2 gives preliminary results on the distribution of $Z$. Formulae on the distribution of $Z$ are given in Section 3 and Appendix A for situation (i). Formulae on the distribution of $Z$ are given in Section 4 and Appendix B for situation (ii). Formulae on the distribution of $Z$ are given in Section 5 and Appendix C for situation (iii). Formulae on the distribution of $Z$ are given in Section 6 and Appendix D for situation (iv). Real data applications of the results in Sections 3–6 are discussed in Section 7. Some conclusions are given in Section 8.

## 2. Preliminary Results

Similarly to [17], we make use of the following assumptions and notations: assume that $(X_1, \ldots, X_p)$ and $(W_1, \ldots, W_p)$ are absolutely continuous; let $F_{X_i}$ denote the cumulative distribution function of $X_i$; let $f_{X_i}$ denote the probability density function of $X_i$; let $F_{W_i}$ denote the cumulative distribution function of $W_i$; let $f_{W_i}$ denote the probability density function of $W_i$; let $H_X(x_1, \ldots, x_p)$ denote the joint cumulative distribution function of $(X_1, \ldots, X_p)$; let $H_W(w_1, \ldots, w_p)$ denote the joint cumulative distribution function of $(W_1, \ldots, W_p)$; let $S_p$ denote the $p$-dimensional simplex defined by

$$S_p = \left\{ (w_1, \ldots, w_p) : w_1 \geq 0, \ldots, w_p \geq 0, w_1 + \cdots + w_p = 1 \right\};$$

let

$$T_p = \left\{ (v_1, \ldots, v_{p-1}) : F_{W_1}^{-1}(v_1) \geq 0, \ldots, F_{W_{p-1}}^{-1}(v_{p-1}) \geq 0, F_{W_1}^{-1}(v_1) + \cdots + F_{W_{p-1}}^{-1}(v_{p-1}) \leq 1 \right\};$$

let

$$A_z = A_{z,p} = \left\{ (x_1, \ldots, x_p) \in R^p : w_1 x_1 + \left( 1 - \sum_{i=1}^{p-1} w_i \right) x_p \leq z \right\};$$

let

$$B_z = B_{z,p} = \left\{ (u_1, \ldots, u_p) \in [0,1]^p : w_1 F_{X_1}^{-1}(u_1) + \cdots + \left( 1 - \sum_{i=1}^{p-1} w_i \right) w_p F_{X_p}^{-1}(u_p) \leq z \right\}.$$

We use the concept of copulas introduced by [31] to describe the dependence among $(X_1, \ldots, X_p)$ and that among $(W_1, \ldots, W_p)$.

**Definition 1.** *A $p$-dimensional function $C : [0,1]^p \to [0,1]$ is a copula if*

$$C(u_1, \ldots, u_{i-1}, 0, u_{i+1}, \ldots, u_p) = 0,$$

$$C(1, \ldots, 1, u, 1, \ldots, 1) = u$$

*and $C$ is $p$ non-decreasing; that is, for every $B = \prod_{i=1}^p [x_i, y_i] \subseteq [0,1]^p$,*

$$\int_B dC(u_1, \ldots, u_p) \geq 0.$$

The fundamental theorem of copulas due to [31] is the following.

**Theorem 1.** *Let H be a p-dimensional cumulative distribution function with marginal distribution functions $F_{Z_i}$, $i = 1, \ldots, p$. Then, there exists a copula C such that*

$$H(z_1, \ldots, z_p) = C\left(F_{Z_1}(z_1), \ldots, F_{Z_p}(z_p)\right).$$

*Conversely, if $F_{Z_i}$, $i = 1, \ldots, p$ are univariate cumulative distribution functions and C is any copula, then H is a p-dimensional cumulative distribution function with marginals $F_{Z_i}$, $i = 1, \ldots, p$. Further, if all of the marginals are continuous, then C is unique.*

Finally, we introduce the two most popular measures of financial risk highlighted in [17]. They are value at risk and expected shortfall, defined by

$$\text{VaR}_p(Z) = F_Z^{-1}(p) = \inf\{z \in \mathbb{R} : F_Z(z) \geq p\}$$

and

$$ES_p(Z) = E\left[Z | Z \leq \text{VaR}_p(Z)\right],$$

respectively.

## 3. Weighted Distribution Results I

In this section, we assume that $X_1, \ldots, X_n$ are dependent random variables; $W_1, \ldots, W_n$ are independent random variables on an $n$-dimensional simplex; $X_1, \ldots, X_n$ are independent of $W_1, \ldots, W_n$. We study the distribution of $Z = W_1 X_1 + \cdots + W_n X_n$.

**Theorem 2.** *With the stated assumptions, the cumulative distribution function of Z can be expressed as*

$$F_Z(z) = \begin{cases} \displaystyle\int_0^1 F_{X_1}\left(\frac{z}{w_1}\right) f_{W_1}(w_1) dw_1, & \text{if } n = 1, \\[2em] \displaystyle\int_{S_n} \int_{B_z} c_X(u_1, \ldots, u_n) du_1 \cdots du_n f_{W_1}(w_1) \cdots f_{W_{n-1}}(w_{n-1}) \\ \qquad\qquad dw_1 \cdots dw_{n-1}, & \text{if } n \geq 2, \end{cases}$$

*where $c_X$ is the density of a copula.*

**Proof.** The case $n = 1$ is obvious. If $n \geq 2$, then we can write

$$\begin{aligned} F_Z(z) &= P(Z \leq z) \\ &= P(W_1 X_1 + \cdots + W_n X_n \leq z) \\ &= P(W_1 X_1 + \cdots + (1 - W_1 - \cdots - W_{n-1}) X_n \leq z) \\ &= \int_{S_n} P\left(W_1 X_1 + \cdots + \left(1 - \sum_{i=1}^{n-1} W_i\right) X_n \leq z \Big| W_1 = w_1, \ldots, W_{n-1} = w_{n-1}\right) \\ &\qquad\qquad f_{W_1}(w_1) \cdots f_{W_{n-1}}(w_{n-1}) dw_1 \cdots dw_{n-1} \\ &= \int_{S_n} P\left(w_1 X_1 + \ldots + \left(1 - \sum_{i=1}^{n-1} w_i\right) X_n \leq z\right) f_{W_1}(w_1) \cdots f_{W_{n-1}}(w_{n-1}) \\ &\qquad\qquad dw_1 \cdots dw_{n-1}, \end{aligned} \tag{1}$$

where the last step follows because of the independence of $X_1, \ldots, X_n$ and $W_1, \ldots, W_n$. Writing

$$P\left(w_1 X_1 + \ldots + \left(1 - \sum_{i=1}^{n-1} w_i\right) X_n \leq z\right) = \int_{A_z} \frac{\partial^n}{\partial x_1 \cdots \partial x_n} H_X(x_1, \ldots, x_n) dx_1 \cdots dx_n,$$

and using Theorem 1, we can rewrite (1) as

$$\int_{S_n} \int_{A_z} \frac{\partial^n}{\partial x_1 \cdots \partial x_n} H_X(x_1, \ldots, x_n) f_{W_1}(w_1) \cdots f_{W_{n-1}}(w_{n-1})$$
$$dx_1 \cdots dx_n dw_1 \cdots dw_{n-1}$$
$$= \int_{S_n} \int_{A_z} \frac{\partial^n}{\partial x_1 \cdots \partial x_n} C_X(F_{X_1}(x_1), \ldots, F_{X_n}(x_n)) f_{W_1}(w_1) \cdots f_{W_{n-1}}(w_{n-1})$$
$$dx_1 \cdots dx_n dw_1 \cdots dw_{n-1}$$
$$= \int_{S_n} \int_{A_z} c_X(F_{X_1}(x_1), \ldots, F_{X_n}(x_n)) f_{X_1}(x_1) \cdots f_{X_n}(x_n)$$
$$f_{W_1}(w_1) \cdots f_{W_{n-1}}(w_{n-1}) dx_1 \cdots dx_n dw_1 \cdots dw_{n-1}. \tag{2}$$

The result follows by substituting $u_1 = F_{X_1}(x_1), \ldots, u_n = F_{X_n}(x_n)$ into (2).   □

**Theorem 3.** *With the stated assumptions, the probability density function of Z can be expressed as*

$$f_Z(z) = \begin{cases} \displaystyle\int_0^1 \frac{1}{w_1} f_{X_1}\left(\frac{z}{w_1}\right) f_{W_1}(w_1) dw_1, & \text{if } n = 1, \\[2em] \displaystyle\int_{S_n} \frac{\partial}{\partial z}\left[\int_{B_z} c_X(u_1, \ldots, u_n) du_1 \cdots du_n\right] \\ \qquad f_{W_1}(w_1) \cdots f_{W_{n-1}}(w_{n-1}) dw_1 \cdots dw_{n-1}, & \text{if } n \geq 2, \end{cases}$$

*where $c_X$ is the density of a copula.*

**Proof.** Immediate from Theorem 2.   □

**Theorem 4.** *With the stated assumptions, the mode of the probability density function of Z is the root of*

$$\int_0^1 \frac{1}{w_1^2} f'_{X_1}\left(\frac{z}{w_1}\right) f_{W_1}(w_1) dw_1 = 0$$

*if $n = 1$ and*

$$\int_{S_n} \frac{\partial^2}{\partial z^2}\left[\int_{B_z} c_X(u_1, \ldots, u_n) du_1 \cdots du_n\right] f_{W_1}(w_1) \cdots f_{W_{n-1}}(w_{n-1}) dw_1 \cdots dw_{n-1} = 0$$

*if $n \geq 2$, where $c_X$ is the density of a copula.*

**Proof.** Immediate from Theorem 3.   □

**Theorem 5.** *With the stated assumptions, the conditional expectation of Z can be expressed as*

$$
E(Z \mid Z \leq z) = \begin{cases} \dfrac{1}{F_Z(z)} \displaystyle\int_0^1 \int_{-\infty}^{\frac{z}{w_1}} w_1 x_1 f_{X_1}(x_1) f_{W_1}(w_1) dx_1 dw_1, & \text{if } n = 1, \\[2em] \dfrac{1}{F_Z(z)} \displaystyle\int_{S_n} \int_{B_z} \left[ w_1 F_{X_1}^{-1}(u_1) + \cdots + \left(1 - \sum_{i=1}^{n-1} w_i\right) F_{X_n}^{-1}(u_n) \right] \\ \qquad c_X(u_1, \ldots, u_n) f_{W_1}(w_1) \cdots f_{W_{n-1}}(w_{n-1}) \\ \qquad du_1 \cdots du_n dw_1 \cdots dw_{n-1}, & \text{if } n \geq 2, \end{cases}
$$

*where $c_X$ is the density of a copula.*

**Proof.** The case $n = 1$ is obvious. If $n \geq 2$, then we can write

$$
E(Z|Z \leq z) = \int_{S_n} E(Z|Z \leq z, W_1 = w_1, \ldots, W_n = w_n) f_{W_1}(w_1) \cdots f_{W_{n-1}}(w_{n-1}) dw_1 \cdots dw_{n-1}
$$

$$
= \int_{S_n} E(W_1 X_1 + \cdots + W_n X_n \mid W_1 X_1 + \cdots + W_n X_n \leq z, W_1 = w_1, \ldots, W_n = w_n)
$$

$$
f_{W_1}(w_1) \cdots f_{W_{n-1}}(w_{n-1}) dw_1 \cdots dw_{n-1}
$$

$$
= \frac{1}{F_Z(z)} \int_{S_n} \int_{A_z} (w_1 x_1 + \cdots + w_n x_n) \frac{\partial^n}{\partial x_1 \cdots \partial x_n} H_X(x_1, \ldots, x_n)
$$

$$
dx_1 \cdots dx_n f_{W_1}(w_1) \cdots f_{W_{n-1}}(w_{n-1}) dw_1 \cdots dw_{n-1}
$$

$$
= \frac{1}{F_Z(z)} \int_{S_n} \int_{A_z} (w_1 x_1 + \cdots + w_n x_n) \frac{\partial^n}{\partial x_1 \cdots \partial x_n} C_X\big(F_{X_1}(x_1), \ldots, F_{X_n}(x_n)\big)
$$

$$
dx_1 \cdots dx_n f_{W_1}(w_1) \cdots f_{W_{n-1}}(w_{n-1}) dw_1 \cdots dw_{n-1}
$$

$$
= \frac{1}{F_Z(z)} \int_{S_n} \int_{A_z} (w_1 x_1 + \cdots + w_n x_n)
$$

$$
c_X\big(F_{X_1}(x_1), \ldots, F_{X_n}(x_n)\big) f_{X_1}(x_1) \cdots f_{X_n}(x_n) dx_1 \cdots dx_n
$$

$$
f_{W_1}(w_1) \cdots f_{W_{n-1}}(w_{n-1}) dw_1 \cdots dw_{n-1}. \tag{3}
$$

The result follows by substituting $u_1 = F_{X_1}(x_1), \ldots, u_n = F_{X_n}(x_n)$ into (3). $\square$

### 4. Weighted Distribution Results II

In this section, we assume that $X_1, \ldots, X_N$ are dependent random variables for a fixed $N$; $W_1, \ldots, W_N$ are independent random variables on an $N$-dimensional simplex for a fixed $N$; $X_1, \ldots, X_N$ are independent of $W_1, \ldots, W_N$ for a fixed $N$; $N$ is a positive-integer-valued random variable independent of $X_1, X_2, \ldots$ and $W_1, W_2, \ldots$. We study the distribution of $Z = W_1 X_1 + \cdots + W_N X_N$.

**Theorem 6.** *With the stated assumptions, the cumulative distribution function of Z can be expressed as*

$$
F_Z(z) = P(N = 1) \int_0^1 F_{X_1}\left(\frac{z}{w_1}\right) f_{W_1}(w_1) dw_1
$$

$$
+ \sum_{n=2}^{\infty} P(N = n) \int_{S_n} \int_{B_{z,n}} c_X(u_1, \ldots, u_n) du_1 \cdots du_n
$$

$$
f_{W_1}(w_1) \cdots f_{W_{n-1}}(w_{n-1}) dw_1 \cdots dw_{n-1},
$$

*where $c_X$ is the density of a copula.*

**Proof.** Write

$$F_Z(z) = P(Z \le z) = \sum_{n=1}^{\infty} P(Z \le z \mid N = n) P(N = n)$$

and use Theorem 2. □

**Theorem 7.** *With the stated assumptions, the probability density function of Z can be expressed as*

$$f_Z(z) = P(N = 1) \int_0^1 \frac{1}{w_1} f_{X_1}\left(\frac{z}{w_1}\right) f_{W_1}(w_1) dw_1$$

$$+ \sum_{n=2}^{\infty} P(N = n) \int_{S_n} \frac{\partial}{\partial z} \left\{ \int_{B_{z,n}} c_X(u_1, \ldots, u_n) du_1 \cdots du_n \right\}$$

$$f_{W_1}(w_1) \cdots f_{W_{n-1}}(w_{n-1}) dw_1 \cdots dw_{n-1},$$

*where $c_X$ is the density of a copula.*

**Proof.** Immediate from Theorem 6. □

**Theorem 8.** *With the stated assumptions, the mode of the probability density function of Z is the root of*

$$P(N = 1) \int_0^1 \frac{1}{w_1^2} f_{X_1}'\left(\frac{z}{w_1}\right) f_{W_1}(w_1) dw_1$$

$$+ \sum_{n=2}^{\infty} P(N = n) \int_{S_n} \frac{\partial^2}{\partial z^2} \left\{ \int_{B_{z,n}} c_X(u_1, \ldots, u_n) du_1 \cdots du_n \right\} f_{W_1}(w_1) \cdots f_{W_{n-1}}(w_{n-1}) dw_1 \cdots dw_{n-1} = 0,$$

*where $c_X$ is the density of a copula.*

**Proof.** Immediate from Theorem 7. □

**Theorem 9.** *With the stated assumptions, the conditional expectation of Z can be expressed as*

$$E(Z \mid Z \le z) = \frac{P(N = 1)}{F_Z(z)} \int_0^1 \int_{-\infty}^{\frac{z}{w_1}} w_1 x_1 f_{X_1}(x_1) f_{W_1}(w_1) dx_1 dw_1$$

$$+ \frac{1}{F_Z(z)} \sum_{n=2}^{\infty} P(N = n) \int_{S_n} \int_{B_{z,n}} \left[ w_1 F_{X_1}^{-1}(u_1) + \cdots + \left(1 - \sum_{i=1}^{n-1} w_i\right) F_{X_n}^{-1}(u_n) \right]$$

$$c_X(u_1, \ldots, u_n) f_{W_1}(w_1) \cdots f_{W_{n-1}}(w_{n-1}) du_1 \cdots du_n dw_1 \cdots dw_{n-1},$$

*where $c_X$ is the density of a copula.*

**Proof.** Write

$$E(Z \mid Z \le z) = \sum_{n=1}^{\infty} E(Z \mid Z \le z, N = n) P(N = n)$$

and use Theorem 5. □

### 5. Weighted Distribution Results III

In this section, we assume that $X_1, \ldots, X_n$ are dependent random variables; $W_1, \ldots, W_n$ are dependent random variables on an $n$-dimensional simplex; $X_1, \ldots, X_n$ are independent of $W_1, \ldots, W_n$. We study the distribution of $Z = W_1 X_1 + \cdots + W_n X_n$.

**Theorem 10.** *With the stated assumptions, the cumulative distribution function of $Z$ can be expressed as*

$$
F_Z(z) = \begin{cases}
\displaystyle\int_0^1 F_{X_1}\left(\frac{z}{w_1}\right) f_{W_1}(w_1) dw_1, & \text{if } n = 1, \\[2em]
\displaystyle\int_{T_n} \int_{B_z} c_X(u_1, \ldots, u_n) c_W(v_1, \ldots, v_{n-1}) du_1 \cdots du_n dv_1 \cdots dv_{n-1}, & \text{if } n \geq 2,
\end{cases}
$$

*where $c_X$ and $c_W$ are densities of copulas.*

**Proof.** The case $n = 1$ is obvious. If $n \geq 2$, then we can write

$$
\begin{aligned}
F_Z(z) &= P(Z \leq z) \\
&= P(W_1 X_1 + \cdots + W_n X_n \leq z) \\
&= P(W_1 X_1 + \cdots + (1 - W_1 - \cdots - W_{n-1}) X_n \leq z) \\
&= \int_{S_n} P\left(W_1 X_1 + \cdots + \left(1 - \sum_{i=1}^{n-1} W_i\right) X_n \leq z \mid W_1 = w_1, \ldots, W_{n-1} = w_{n-1}\right) \\
&\qquad \frac{\partial^n}{\partial w_1 \cdots \partial w_{n-1}} H_W(w_1, \ldots, w_{n-1}) dw_1 \cdots dw_{n-1} \\
&= \int_{S_n} P\left(W_1 X_1 + \cdots + \left(1 - \sum_{i=1}^{n-1} W_i\right) X_n \leq z \mid W_1 = w_1, \ldots, W_{n-1} = w_{n-1}\right) \\
&\qquad \frac{\partial^n}{\partial w_1 \cdots \partial w_{n-1}} C_W\left(F_{W_1}(w_1), \ldots, F_{W_{n-1}}(w_{n-1})\right) dw_1 \cdots dw_{n-1} \\
&= \int_{S_n} P\left(W_1 X_1 + \cdots + \left(1 - \sum_{i=1}^{n-1} W_i\right) X_n \leq z \mid W_1 = w_1, \ldots, W_{n-1} = w_{n-1}\right) \\
&\qquad c_W\left(F_{W_1}(w_1), \ldots, F_{W_{n-1}}(w_{n-1})\right) f_{W_1}(w_1) \cdots f_{W_{n-1}}(w_{n-1}) dw_1 \cdots dw_{n-1} \\
&= \int_{S_n} \int_{A_z} \frac{\partial^n}{\partial x_1 \cdots \partial x_n} H_X(x_1, \ldots, x_n) \\
&\qquad c_W\left(F_{W_1}(w_1), \ldots, F_{W_{n-1}}(w_{n-1})\right) f_{W_1}(w_1) \cdots f_{W_{n-1}}(w_{n-1}) \\
&\qquad dx_1 \cdots dx_n dw_1 \cdots dw_{n-1} \\
&= \int_{S_n} \int_{A_z} \frac{\partial^n}{\partial x_1 \cdots \partial x_n} C_X\left(F_{X_1}(x_1), \ldots, F_{X_n}(x_n)\right) \\
&\qquad c_W\left(F_{W_1}(w_1), \ldots, F_{W_{n-1}}(w_{n-1})\right) f_{W_1}(w_1) \cdots f_{W_{n-1}}(w_{n-1}) \\
&\qquad dx_1 \cdots dx_n dw_1 \cdots dw_{n-1} \\
&= \int_{S_n} \int_{A_z} c_X\left(F_{X_1}(x_1), \ldots, F_{X_n}(x_n)\right) f_{X_1}(x_1) \cdots f_{X_n}(x_n) \\
&\qquad c_W\left(F_{W_1}(w_1), \ldots, F_{W_{n-1}}(w_{n-1})\right) f_{W_1}(w_1) \cdots f_{W_{n-1}}(w_{n-1}) \\
&\qquad dx_1 \cdots dx_n dw_1 \cdots dw_{n-1}.
\end{aligned}
\tag{4}
$$

The result follows by substituting $u_1 = F_{X_1}(x_1), \ldots, u_n = F_{X_n}(x_n)$ and $v_1 = F_{W_1}(w_1), \ldots, v_{n-1} = F_{W_{n-1}}(w_{n-1})$ into (4). $\square$

**Theorem 11.** *With the stated assumptions, the probability density function of Z can be expressed as*

$$
f_Z(z) = \begin{cases} \displaystyle \int_0^1 \frac{1}{w_1} f_{X_1}\left(\frac{z}{w_1}\right) f_{W_1}(w_1) dw_1, & \text{if } n = 1, \\[3mm] \displaystyle \int_{T_n} \frac{\partial}{\partial z} \left[ \int_{B_z} c_X(u_1, \ldots, u_n) du_1 \cdots du_n \right] c_W(v_1, \ldots, v_{n-1}) dv_1 \cdots dv_{n-1}, & \text{if } n \geq 2, \end{cases}
$$

*where $c_X$ and $c_W$ are densities of copulas.*

**Proof.** Immediate from Theorem 10. $\square$

**Theorem 12.** *With the stated assumptions, the mode of the probability density function of Z is the root of*

$$
\int_0^1 \frac{1}{w_1^2} f'_{X_1}\left(\frac{z}{w_1}\right) f_{W_1}(w_1) dw_1 = 0
$$

*if $n = 1$ and*

$$
\int_{T_n} \frac{\partial^2}{\partial z^2} \left[ \int_{B_z} c_X(u_1, \ldots, u_n) du_1 \cdots du_n \right] c_W(v_1, \ldots, v_{n-1}) dv_1 \cdots dv_{n-1} = 0
$$

*if $n \geq 2$, where $c_X$ and $c_W$ are densities of copulas.*

**Proof.** Immediate from Theorem 11. $\square$

**Theorem 13.** *With the stated assumptions, the conditional expectation of Z can be expressed as*

$$
E(Z \mid Z \leq z) = \begin{cases} \displaystyle \frac{1}{F_Z(z)} \int_0^1 \int_{-\infty}^{\frac{z}{w_1}} w_1 x_1 f_{X_1}(x_1) f_{W_1}(w_1) dx_1 dw_1, & \text{if } n = 1, \\[4mm] \displaystyle \frac{1}{F_Z(z)} \int_{T_n} \int_{B_z} \left[ F_{W_1}^{-1}(v_1) F_{X_1}^{-1}(u_1) + \cdots + \left(1 - \sum_{i=1}^{n-1} F_{W_i}^{-1}(v_i)\right) F_{X_n}^{-1}(u_n) \right] \\ \qquad c_X(u_1, \ldots, u_n) c_W(v_1, \ldots, v_{n-1}) du_1 \cdots du_n dv_1 \cdots dv_{n-1}, \\ \qquad \text{if } n \geq 2, \end{cases}
$$

*where $c_X$ and $c_W$ are densities of copulas.*

**Proof.** The case $n = 1$ is obvious. If $n \geq 2$, then we can write

$$E(Z|Z \leq z) = \int_{S_n} E(Z|Z \leq z, W_1 = w_1, \ldots, W_n = w_n) \frac{\partial^{n-1}}{\partial w_1 \cdots \partial w_{n-1}} H_W(w_1, \ldots, w_{n-1})$$
$$dw_1 \cdots dw_{n-1}$$

$$= \int_{S_n} E(W_1 X_1 + \cdots + W_n X_n \mid W_1 X_1 + \cdots + W_n X_n \leq z, W_1 = w_1, \ldots, W_n = w_n)$$
$$\frac{\partial^{n-1}}{\partial w_1 \cdots \partial w_{n-1}} H_W(w_1, \ldots, w_{n-1}) dw_1 \cdots dw_{n-1}$$

$$= \int_{S_n} E(W_1 X_1 + \cdots + W_n X_n \mid W_1 X_1 + \cdots + W_n X_n \leq z, W_1 = w_1, \ldots, W_n = w_n)$$
$$\frac{\partial^{n-1}}{\partial w_1 \cdots \partial w_{n-1}} C_W\big(F_{W_1}(w_1), \ldots, F_{W_{n-1}}(w_{n-1})\big) dw_1 \cdots dw_{n-1}$$

$$= \int_{S_n} E(W_1 X_1 + \cdots + W_n X_n \mid W_1 X_1 + \cdots + W_n X_n \leq z, W_1 = w_1, \ldots, W_n = w_n)$$
$$c_W\big(F_{W_1}(w_1), \ldots, F_{W_{n-1}}(w_{n-1})\big) f_{W_1}(w_1) \cdots f_{W_{n-1}}(w_{n-1})$$
$$dw_1 \cdots dw_{n-1}$$

$$= \frac{1}{F_Z(z)} \int_{S_n} \int_{A_z} (w_1 x_1 + \cdots + w_n x_n)$$
$$\frac{\partial^n}{\partial x_1 \cdots \partial x_n} H_X(x_1, \ldots, x_n) dx_1 \cdots dx_n$$
$$c_W\big(F_{W_1}(w_1), \ldots, F_{W_{n-1}}(w_{n-1})\big) f_{W_1}(w_1) \cdots f_{W_{n-1}}(w_{n-1})$$
$$dw_1 \cdots dw_{n-1}$$

$$= \frac{1}{F_Z(z)} \int_{S_n} \int_{A_z} (w_1 x_1 + \cdots + w_n x_n)$$
$$\frac{\partial^n}{\partial x_1 \cdots \partial x_n} C_X\big(F_{X_1}(x_1), \ldots, F_{X_n}(x_n)\big) dx_1 \cdots dx_n$$
$$c_W\big(F_{W_1}(w_1), \ldots, F_{W_{n-1}}(w_{n-1})\big) f_{W_1}(w_1) \cdots f_{W_{n-1}}(w_{n-1})$$
$$dw_1 \cdots dw_{n-1}$$

$$= \frac{1}{F_Z(z)} \int_{S_n} \int_{A_z} (w_1 x_1 + \cdots + w_n x_n)$$
$$c_X\big(F_{X_1}(x_1), \ldots, F_{X_n}(x_n)\big) f_{X_1}(x_1) \cdots f_{X_n}(x_n) dx_1 \cdots dx_n$$
$$c_W\big(F_{W_1}(w_1), \ldots, F_{W_{n-1}}(w_{n-1})\big) f_{W_1}(w_1) \cdots f_{W_{n-1}}(w_{n-1})$$
$$dw_1 \cdots dw_{n-1}. \tag{5}$$

The result follows by substituting $u_1 = F_{X_1}(x_1), \ldots, u_n = F_{X_n}(x_n)$ and $v_1 = F_{W_1}(w_1), \ldots, v_{n-1} = F_{W_{n-1}}(w_{n-1})$ into (5). $\square$

## 6. Weighted Distribution Results IV

In this section, we assume that $X_1, \ldots, X_N$ are dependent random variables for a fixed $N$; $W_1, \ldots, W_N$ are dependent random variables on an $N$-dimensional simplex for a fixed $N$; $X_1, \ldots, X_N$ are independent of $W_1, \ldots, W_N$ for a fixed $N$; $N$ is a positive-integer-valued random variable independent of $X_1, X_2, \ldots$ and $W_1, W_2, \ldots$. We study the distribution of $Z = W_1 X_1 + \cdots + W_N X_N$.

**Theorem 14.** *With the stated assumptions, the cumulative distribution function of Z can be expressed as*

$$F_Z(z) = P(N=1) \int_0^1 F_{X_1}\left(\frac{z}{w_1}\right) f_{W_1}(w_1) dw_1$$

$$+ \sum_{n=2}^{\infty} P(N=n) \int_{T_n} \int_{B_{z,n}} c_X(u_1, \ldots, u_n) c_W(v_1, \ldots, v_{n-1}) du_1 \cdots du_n dv_1 \cdots dv_{n-1},$$

*where $c_X$ and $c_W$ are densities of copulas.*

**Proof.** Write

$$F_Z(z) = P(Z \leq z) = \sum_{n=1}^{\infty} P(Z \leq z \mid N=n) P(N=n)$$

and use Theorem 10. □

**Theorem 15.** *With the stated assumptions, the probability density function of Z can be expressed as*

$$f_Z(z) = P(N=1) \int_0^1 \frac{1}{w_1} f_{X_1}\left(\frac{z}{w_1}\right) f_{W_1}(w_1) dw_1$$

$$+ \sum_{n=2}^{\infty} P(N=n) \int_{T_n} \frac{\partial}{\partial z} \left\{ \int_{B_{z,n}} c_X(u_1, \ldots, u_n) du_1 \cdots du_n \right\} c_W(v_1, \ldots, v_{n-1}) dv_1 \cdots dv_{n-1},$$

*where $c_X$ and $c_W$ are densities of copulas.*

**Proof.** Immediate from Theorem 14. □

**Theorem 16.** *With the stated assumptions, the mode of the probability density function of Z is the root of*

$$P(N=1) \int_0^1 \frac{1}{w_1^2} f'_{X_1}\left(\frac{z}{w_1}\right) f_{W_1}(w_1) dw_1$$

$$+ \sum_{n=2}^{\infty} P(N=n) \int_{T_n} \frac{\partial^2}{\partial z^2} \left\{ \int_{B_{z,n}} c_X(u_1, \ldots, u_n) du_1 \cdots du_n \right\} c_W(v_1, \ldots, v_{n-1}) dv_1 \cdots dv_{n-1} = 0,$$

*where $c_X$ and $c_W$ are densities of copulas.*

**Proof.** Immediate from Theorem 15. □

**Theorem 17.** *With the stated assumptions, the conditional expectation of Z can be expressed as*

$$E(Z \mid Z \leq z) = \frac{P(N=1)}{F_Z(z)} \int_0^1 \int_{-\infty}^{\frac{z}{w_1}} w_1 x_1 f_{X_1}(x_1) f_{W_1}(w_1) dx_1 dw_1$$

$$+ \frac{1}{F_Z(z)} \sum_{n=2}^{\infty} P(N=n) \int_{T_n} \int_{B_{z,n}} \left[ F_{W_1}^{-1}(v_1) F_{X_1}^{-1}(u_1) + \cdots + \left( 1 - \sum_{i=1}^{n-1} F_{W_i}^{-1}(v_i) \right) F_{X_n}^{-1}(u_n) \right]$$

$$c_X(u_1, \ldots, u_n) c_W(v_1, \ldots, v_{n-1}) du_1 \cdots du_n dv_1 \cdots dv_{n-1},$$

*where $c_X$ and $c_W$ are densities of copulas.*

**Proof.** Write

$$E(Z \mid Z \leq z) = \sum_{n=1}^{\infty} E(Z \mid Z \leq z, N=n) P(N=n)$$

and use Theorem 13.  □

## 7. Real Data Applications

In this section, we discuss real examples where the results of Sections 3–6 can be applied. Sections 7.1 and 7.3 use data on the gross domestic product and population of various countries from 1960 to 2022. These data were obtained from the World Bank. We could not obtain real data to illustrate the situations described in Sections 7.2 and 7.4.

### 7.1. Weighted Distribution I

Consider the five largest economics in the world: the United States of America, China, Japan, Germany and India. Let $X_1 =$ gross domestic product for the United States of America, $X_2 =$ gross domestic product for China, $X_3 =$ gross domestic product for Japan, $X_4 =$ gross domestic product for Germany, $X_5 =$ gross domestic product for India, $W_1 =$ population of the United States of America/total population of the five largest economies, $W_2 =$ population of China/total population of the five largest economies, $W_3 =$ population of Japan/total population of the five largest economies, $W_4 =$ population of Germany/total population of the five largest economies and $W_5 =$ population of India/total population of the five largest economies. Then, $Z = W_1 X_1 + \cdots + W_5 X_5$ will denote the gross domestic product per person in the five largest economies. In this case, $(X_1, \ldots, X_5)$ are dependent because of the trade among the largest economies. $(W_1, \ldots, W_4)$ are not likely to be dependent because there is not so much immigration between India and China or India and Japan.

The marginal distributions of $(X_1, \ldots, X_5)$ and $(W_1, \ldots, W_5)$ were modeled using non-parametric kernel densities. The joint distribution of $(X_1, \ldots, X_5)$ was modeled using the Clayton copula [32]. Figure 1 shows the fitted quantile function of $Z$.

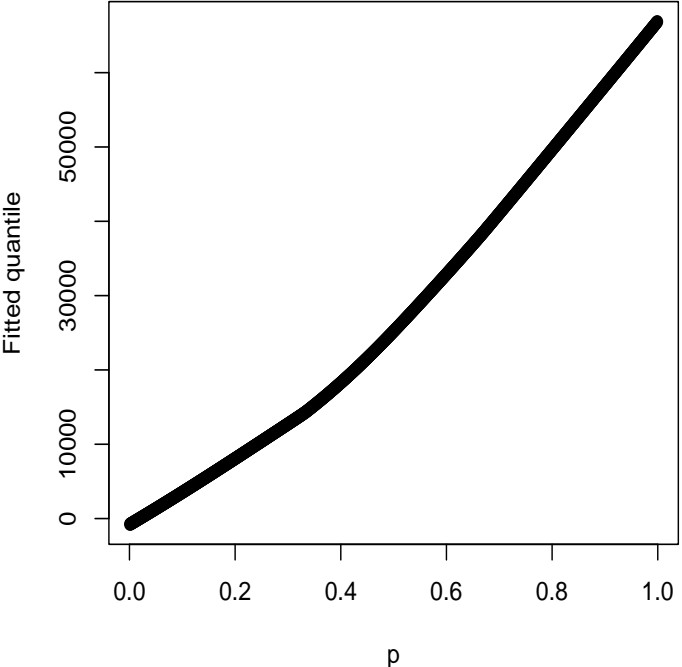

**Figure 1.** Fitting quantile function of $Z = W_1 X_1 + \cdots + W_5 X_5$ versus $p$.

### 7.2. Weighted Distribution II

Suppose that there are $N$ accidents at a busy intersection on a given day. Clearly, $N$ is a random variable. Possible models for $N$ could include the Poisson distribution. Let $X_1, \ldots, X_N$ denote the damages caused by the $N$ accidents. Some of the accidents could involve more than one individual or more than one vehicle. Let $W_1, \ldots, W_N$ denote scale factors corresponding to one individual or one vehicle. Then, $Z = W_1 X_1 + \cdots + W_N X_N$ will

denote the average damage for the individual or vehicle. In this case, $(X_1, \ldots, X_N)$ given $N$ could be dependent because of the possibility of serial accidents. But, $(W_1, \ldots, W_{N-1})$ given $N$ are likely to be independent.

### 7.3. Weighted Distribution III

Consider the countries in the Indian subcontinent: India, Nepal, Bangladesh, Pakistan, Bhutan, Sri Lanka and the Maldives. Let $X_1$ = gross domestic product for India, $X_2$ = gross domestic product for Nepal, $X_3$ = gross domestic product for Bangladesh, $X_4$ = gross domestic product for Pakistan, $X_5$ = gross domestic product for Bhutan, $X_6$ = gross domestic product for Sri Lanka, $X_7$ = gross domestic product for the Maldives, $W_1$ = population of India/total population of the Indian subcontinent, $W_2$ = population of Nepal/total population of the Indian subcontinent, $W_3$ = population of Bangladesh/total population of the Indian subcontinent, $W_4$ = population of Pakistan/total population of the Indian subcontinent, $W_5$ = population of Bhutan/total population of the Indian subcontinent, $W_6$ = population of Sri Lanka/total population of the Indian subcontinent and $W_7$ = population of the Maldives/total population of the Indian subcontinent. Then, $Z = W_1 X_1 + \cdots + W_7 X_7$ will denote the gross domestic product per person in the Indian subcontinent. In this case, $(X_1, \ldots, X_7)$ are dependent because of the trade among the countries. Likewise, $(W_1, \ldots, W_6)$ are dependent because of immigration among neighboring countries.

The marginal distributions of $(X_1, \ldots, X_7)$ and $(W_1, \ldots, W_7)$ were modeled using nonparametric kernel densities. The joint distribution of $(X_1, \ldots, X_7)$ and the joint distribution of $(W_1, \ldots, W_7)$ were modeled using the Clayton copula. Figure 2 shows the fitted quantile function of $Z$. There appears to be a shift at around $p = 0.5$.

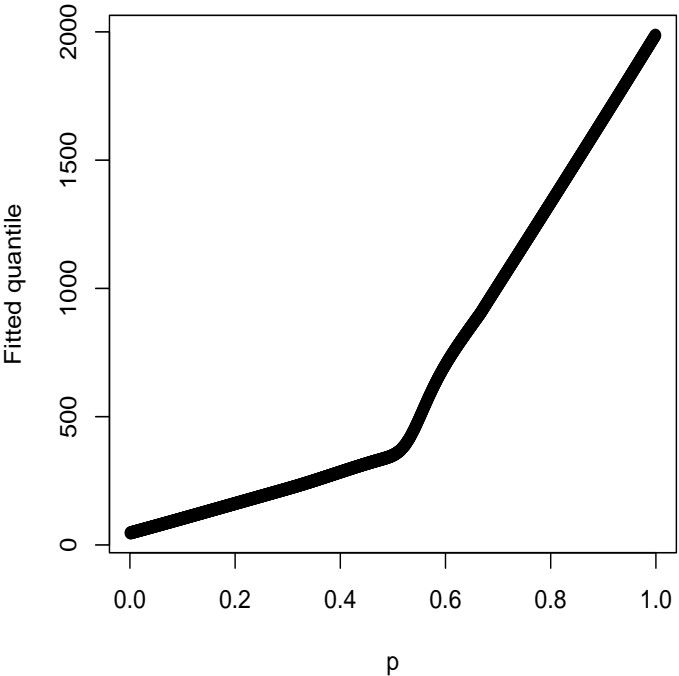

**Figure 2.** Fitting quantile function of $Z = W_1 X_1 + \cdots + W_7 X_7$ versus $p$.

### 7.4. Weighted Distribution IV

Let $N$ denote the number of goals scored in a football match. Let $X_i$, $i = 1, \ldots, N$ denote the amount paid out by a betting company if $i$ goals were scored. Let $W_i$, $i = 1, \ldots, N$ denote the odds of there being $i$ goals scored determined by the betting company. Then, $Z = W_1 X_1 + \cdots + W_N X_N$ will denote the average amount paid out by the company. In this case, $(X_1, \ldots, X_N)$ will be dependent and so will $(W_1, \ldots, W_{N-1})$.

## 8. Conclusions

Let $X_1, \ldots, X_n$ be random variables and let $W_1, \ldots, W_n$ be random weights, independent of $X_1, \ldots, X_n$ taking values on the unit simplex. Let $Z = W_1 X_1 + \cdots + W_n X_n$. In this paper, we have derived formulae for the cumulative distribution function, probability density function, conditional expectation, moment generating function, characteristic function, cumulant generating function, $m$th moment, value at risk and expected shortfall of $Z$, considering four possible situations. We have discussed real situations where these formulas can be applied. Future work is to derive exact results for the distribution of $\mathbf{Z} = \mathbf{W}_1 \mathbf{X}_1 + \cdots + (\mathbf{I} - \mathbf{W}_1 - \cdots - \mathbf{W}_{n-1}) \mathbf{X}_n$, where $\mathbf{X}_1, \ldots, \mathbf{X}_n$ are dependent multivariate random variables.

**Author Contributions:** Both authors contributed equally to the manuscript. All authors have read and agreed to the published version of the manuscript.

**Funding:** This research received no external funding.

**Data Availability Statement:** The data can be obtained from the corresponding author.

**Acknowledgments:** The authors would like to thank the editor and the three referees for careful reading and comments that greatly improved the paper.

**Conflicts of Interest:** The authors declare no conflict of interest.

## Appendix A. Supplementary Results for Weighted Distribution I

**Theorem A1.** *With the assumptions stated in Section 3, the moment generating function of $Z$ can be expressed as*

$$
M_Z(t) = \begin{cases} \displaystyle\int_0^1 \frac{1}{w_1} \int_{-\infty}^{\infty} \exp(tz) f_{X_1}\left(\frac{z}{w_1}\right) f_{W_1}(w_1)\,dz\,dw_1, & \text{if } n = 1, \\[2em] \displaystyle\int_{S_n} f_{W_1}(w_1) \cdots f_{W_{n-1}}(w_{n-1}) \\ \qquad\qquad \displaystyle\int_{-\infty}^{\infty} \exp(tz)\left\{ \frac{\partial}{\partial z} \int_{B_z} c_X(u_1 \ldots, u_n)\,du_1 \cdots du_n \right\} \\ \qquad\qquad dz\,dw_1 \cdots dw_{n-1}, & \text{if } n \geq 2, \end{cases}
$$

*where $c_X$ is the density of a copula.*

**Proof.** The result follows from

$$
M_Z(t) = E[\exp(tZ)] = \int_{-\infty}^{\infty} \exp(tz) f_Z(z)\,dz
$$

and Theorem 3. $\square$

**Theorem A2.** *With the assumptions stated in Section 3, the characteristic function of $Z$ can be expressed as*

$$
\phi_Z(t) = \begin{cases} \displaystyle\int_0^1 \frac{1}{w_1} \int_{-\infty}^{\infty} \exp(itz) f_{X_1}\left(\frac{z}{w_1}\right) f_{W_1}(w_1)\,dz\,dw_1, & \text{if } n = 1, \\[2em] \displaystyle\int_{S_n} f_{W_1}(w_1) \cdots f_{W_{n-1}}(w_{n-1}) \\ \qquad\qquad \displaystyle\int_{-\infty}^{\infty} \exp(itz)\left\{ \frac{\partial}{\partial z} \int_{B_z} c_X(u_1 \ldots, u_n)\,du_1 \cdots du_n \right\} \\ \qquad\qquad dz\,dw_1 \cdots dw_{n-1}, & \text{if } n \geq 2, \end{cases}
$$

*where* $i = \sqrt{-1}$ *and* $c_X$ *is the density of a copula.*

**Proof.** The result follows from

$$\phi_Z(t) = E[\exp(itZ)] = \int_{-\infty}^{\infty} \exp(itz) f_Z(z) dz$$

and Theorem 3. □

**Theorem A3.** *With the assumptions stated in Section 3, the cumulant generating function of Z can be expressed as*

$$K_Z(t) = \begin{cases} \log\left\{ \int_0^1 \frac{1}{w_1} \int_{-\infty}^{\infty} \exp(itz) f_{X_1}\left(\frac{z}{w_1}\right) f_{W_1}(w_1) dz dw_1 \right\}, & if\ n = 1, \\[2em] \log\left\{ \int_{S_n} f_{W_1}(w_1) \cdots f_{W_{n-1}}(w_{n-1}) \right. \\ \qquad \int_{-\infty}^{\infty} \exp(itz) \left\{ \frac{\partial}{\partial z} \int_{B_z} c_X(u_1 \ldots, u_n) dz du_1 \cdots du_n \right\} \\ \left. dw_1 \cdots dw_{n-1} \right\}, & if\ n \geq 2, \end{cases}$$

*where* $i = \sqrt{-1}$ *and* $c_X$ *is the density of a copula.*

**Proof.** Immediate from Theorem A2. □

**Theorem A4.** *With the assumptions stated in Section 3, the mth moment of Z can be expressed as*

$$E(Z^m) = \begin{cases} \int_0^1 \frac{1}{w_1} \int_{-\infty}^{\infty} z^m f_{X_1}\left(\frac{z}{w_1}\right) f_{W_1}(w_1) dz dw_1, & if\ n = 1, \\[2em] \int_{S_n} f_{W_1}(w_1) \cdots f_{W_{n-1}}(w_{n-1}) \\ \qquad \int_{-\infty}^{\infty} z^m \left\{ \frac{\partial}{\partial z} \int_{B_z} c_X(u_1 \ldots, u_n) dz du_1 \cdots du_n \right\} \\ \qquad dw_1 \cdots dw_{n-1}, & if\ n \geq 2, \end{cases}$$

*where* $c_X$ *is the density of a copula.*

**Proof.** The result follows from

$$E(Z^m) = \int_{-\infty}^{\infty} z^m f_Z(z) dz$$

and Theorem 3. □

**Corollary A1.** *With the assumptions stated in Section 3, the variance, skewness and kurtosis of Z can be computed using the relationships*

$$Var(Z) = E\left[Z^2\right] - \{E[Z]\}^2,$$

$$Skewness(Z) = \frac{E\left[Z^3\right] - 3\{E[Z]\}^2 E[Z] - \{E[Z]\}^3}{\left[E[Z^2] - \{E[Z]\}^2\right]^{\frac{3}{2}}}$$

*and*

$$Kurtosis(Z) = \frac{E[Z^4] - 4E[Z^3]E[Z] + 6E[Z^2]\{E[Z]\}^2 - 3\{E[Z]\}^4}{\left[E[Z^2] - \{E[Z]\}^2\right]^2}.$$

**Corollary A2.** *With the assumptions stated in Section 3, the mth cumulant of Z can be computed using*

$$\kappa_m = K_Z^{(m)}(0) = \left. \frac{d^m K_Z(t)}{dt^m} \right|_{t=0}.$$

**Theorem A5.** *With the assumptions stated in Section 3, the value at risk of Z with probability p can be expressed as the root of*

$$\int_0^1 F_{X_1}\left(\frac{\mathrm{VaR}_Z(p)}{w_1}\right) f_{W_1}(w_1)dw_1 = p \tag{A1}$$

*if $n = 1$ and as the root of*

$$\int_{S_n} \int_{B_{\mathrm{VaR}_Z(p)}} c_X(u_1,\ldots,u_n)du_1\cdots du_n f_{W_1}(w_1)\cdots f_{W_{n-1}}(w_{n-1})dw_1\cdots dw_{n-1} = p \tag{A2}$$

*if $n \geq 2$, where $c_X$ is the density of a copula.*

**Proof.** (A1) and (A2) follow immediately from Theorem 2. □

**Theorem A6.** *With the assumptions stated in Section 3, the expected shortfall of Z with probability p can be expressed as*

$$\mathrm{ES}_p(Z) = \begin{cases} \dfrac{1}{p} \displaystyle\int_0^1 \int_{-\infty}^{\frac{\mathrm{VaR}_p(Z)}{w_1}} w_1 x_1 f_{X_1}(x_1) f_{W_1}(w_1)dx_1 dw_1, & \text{if } n = 1, \\[20pt] \dfrac{1}{p} \displaystyle\int_{S_n} \int_{B_{\mathrm{VaR}_Z(p)}} \left[ w_1 F_{X_1}^{-1}(u_1) + \cdots + \left(1 - \sum_{i=1}^{n-1} w_i\right) F_{X_n}^{-1}(u_n) \right] \\ \qquad c_X(u_1,\ldots,u_n) f_{W_1}(w_1)\cdots f_{W_{n-1}}(w_{n-1}) \\ \qquad du_1\cdots du_n dw_1\cdots dw_{n-1}, & \text{if } n \geq 2, \end{cases}$$

*where $c_X$ is the density of a copula.*

**Proof.** (A3) follows immediately from Theorem 5. □

## Appendix B. Supplementary Results for Weighted Distribution II

**Theorem A7.** *With the assumptions stated in Section 4, the moment generating function of Z can be expressed as*

$$M_Z(t) = P(N = 1) \int_0^1 \frac{1}{w_1} \int_{-\infty}^{\infty} \exp(tz) f_{X_1}\left(\frac{z}{w_1}\right) f_{W_1}(w_1)dzdw_1$$

$$+ \sum_{n=2}^{\infty} P(N = n) \int_{S_n} \int_{-\infty}^{\infty} \exp(tz) \frac{\partial}{\partial z}\left\{ \int_{B_{z,n}} c_X(u_1,\ldots,u_n)du_1\cdots du_n \right\}$$

$$f_{W_1}(w_1)\cdots f_{W_{n-1}}(w_{n-1})dzdw_1\cdots dw_{n-1},$$

*where $c_X$ is the density of a copula.*

**Proof.** The result follows from

$$M_Z(t) = E[\exp(tZ)] = \int_{-\infty}^{\infty} \exp(tz) f_Z(z) dz$$

and Theorem 7. □

**Theorem A8.** *With the assumptions stated in Section 4, the characteristic function of Z can be expressed as*

$$\phi_Z(t) = P(N = 1) \int_0^1 \frac{1}{w_1} \int_{-\infty}^{\infty} \exp(\mathrm{i}tz) f_{X_1}\left(\frac{z}{w_1}\right) f_{W_1}(w_1) dz dw_1$$
$$+ \sum_{n=2}^{\infty} P(N = n) \int_{S_n} \int_{-\infty}^{\infty} \exp(\mathrm{i}tz) \frac{\partial}{\partial z} \left\{ \int_{B_{z,n}} c_X(u_1, \ldots, u_n) du_1 \cdots du_n \right\}$$
$$f_{W_1}(w_1) \cdots f_{W_{n-1}}(w_{n-1}) dz dw_1 \cdots dw_{n-1},$$

*where* $\mathrm{i} = \sqrt{-1}$ *and* $c_X$ *is the density of a copula.*

**Proof.** The result follows from

$$\phi_Z(t) = E[\exp(\mathrm{i}tZ)] = \int_{-\infty}^{\infty} \exp(\mathrm{i}tz) f_Z(z) dz$$

and Theorem 7. □

**Theorem A9.** *With the assumptions stated in Section 4, the cumulant generating function of Z can be expressed as*

$$K_Z(t) = \log \left\{ P(N = 1) \int_0^1 \frac{1}{w_1} \int_{-\infty}^{\infty} \exp(\mathrm{i}tz) f_{X_1}\left(\frac{z}{w_1}\right) f_{W_1}(w_1) dz dw_1 \right.$$
$$+ \sum_{n=2}^{\infty} P(N = n) \int_{S_n} \int_{-\infty}^{\infty} \exp(\mathrm{i}tz) \frac{\partial}{\partial z} \left\{ \int_{B_{z,n}} c_X(u_1, \ldots, u_n) du_1 \cdots du_n \right\}$$
$$\left. f_{W_1}(w_1) \cdots f_{W_{n-1}}(w_{n-1}) dz dw_1 \cdots dw_{n-1} \right\},$$

*where* $\mathrm{i} = \sqrt{-1}$ *and* $c_X$ *is the density of a copula.*

**Proof.** Immediate from Theorem A8. □

**Theorem A10.** *With the assumptions stated in Section 4, the mth moment of Z can be expressed as*

$$E(Z^m) = P(N = 1) \int_0^1 \frac{1}{w_1} \int_{-\infty}^{\infty} z^m f_{X_1}\left(\frac{z}{w_1}\right) f_{W_1}(w_1) dz dw_1$$
$$+ \sum_{n=2}^{\infty} P(N = n) \int_{S_n} \int_{-\infty}^{\infty} z^m \frac{\partial}{\partial z} \left\{ \int_{B_{z,n}} c_X(u_1, \ldots, u_n) du_1 \cdots du_n \right\}$$
$$f_{W_1}(w_1) \cdots f_{W_{n-1}}(w_{n-1}) dz dw_1 \cdots dw_{n-1},$$

*where* $c_X$ *is the density of a copula.*

**Proof.** The result follows from

$$E(Z^m) = \int_{-\infty}^{\infty} z^m f_Z(z) dz$$

and Theorem 7. □

**Theorem A11.** *With the assumptions stated in Section 4, the value at risk of Z with probability p can be expressed as the root of*

$$P(N = 1) \int_0^1 F_{X_1}\left(\frac{\text{VaR}_Z(p)}{w_1}\right) f_{W_1}(w_1) dw_1$$

$$+ \sum_{n=2}^{\infty} P(N = n) \int_{S_n} \int_{B_{\text{VaR}_Z(p),n}} c_X(u_1, \ldots, u_n) du_1 \cdots du_n$$

$$f_{W_1}(w_1) \cdots f_{W_{n-1}}(w_{n-1}) dw_1 \cdots dw_{n-1} = p, \quad (A3)$$

*where $c_X$ is the density of a copula.*

**Proof.** (A3) follows immediately from Theorem 6. □

**Theorem A12.** *With the assumptions stated in Section 4, the expected shortfall of Z with probability p can be expressed as*

$$\text{ES}_p(Z) = \frac{P(N = 1)}{p} \int_0^1 \int_{-\infty}^{\frac{\text{VaR}_p(Z)}{w_1}} w_1 x_1 f_{X_1}(x_1) f_{W_1}(w_1) dx_1 dw_1$$

$$+ \frac{1}{p} \sum_{n=2}^{\infty} P(N = n) \int_{S_n} \int_{B_{\text{VaR}_Z(p),n}} \left[ w_1 F_{X_1}^{-1}(u_1) + \cdots + \left(1 - \sum_{i=1}^{n-1} w_i\right) F_{X_n}^{-1}(u_n) \right]$$

$$c_X(u_1, \ldots, u_n) f_{W_1}(w_1) \cdots f_{W_{n-1}}(w_{n-1}) du_1 \cdots du_n dw_1 \cdots dw_{n-1}, \quad (A4)$$

*where $c_X$ is the density of a copula.*

**Proof.** (A4) follows immediately from Theorem 9. □

**Appendix C. Supplementary Results for Weighted Distribution III**

**Theorem A13.** *With the stated assumptions in Section 5, the moment generating function of Z can be expressed as*

$$M_Z(t) = \begin{cases} \int_0^1 \frac{1}{w_1} \int_{-\infty}^{\infty} \exp(tz) f_{X_1}\left(\frac{z}{w_1}\right) f_{W_1}(w_1) dz dw_1, & \text{if } n = 1, \\ \\ \int_{T_n} \int_{-\infty}^{\infty} \exp(tz) \frac{\partial}{\partial z} \left[ \int_{B_z} c_X(u_1, \ldots, u_n) du_1 \cdots du_n \right] dz c_W(v_1, \ldots, v_{n-1}) dv_1 \cdots dv_{n-1}, \\ \quad \text{if } n \geq 2, \end{cases}$$

*where $c_X$ and $c_W$ are densities of copulas.*

**Proof.** The result follows from

$$M_Z(t) = E[\exp(tZ)] = \int_{-\infty}^{\infty} \exp(tz) f_Z(z) dz$$

and Theorem 11. □

**Theorem A14.** *With the stated assumptions in Section 5, the characteristic function of Z can be expressed as*

$$\phi_Z(t) = \begin{cases} \int_{-\infty}^{\infty} \exp(\mathrm{i}tz) \int_0^1 \frac{1}{w_1} f_{X_1}\left(\frac{z}{w_1}\right) f_{W_1}(w_1) dw_1 dz, & \text{if } n = 1, \\[2em] \int_{T_n} \int_{-\infty}^{\infty} \exp(\mathrm{i}tz) \frac{\partial}{\partial z} \left[ \int_{B_z} c_X(u_1, \ldots, u_n) du_1 \cdots du_n \right] dz c_W(v_1, \ldots, v_{n-1}) dv_1 \cdots dv_{n-1}, \\ \quad \text{if } n \geq 2, \end{cases}$$

*where* $\mathrm{i} = \sqrt{-1}$ *and* $c_X$ *and* $c_W$ *are densities of copulas.*

**Proof.** The result follows from

$$\phi_Z(t) = E[\exp(\mathrm{i}tZ)] = \int_{-\infty}^{\infty} \exp(\mathrm{i}tz) f_Z(z) dz$$

and Theorem 11. $\quad\square$

**Theorem A15.** *With the stated assumptions in Section 5, the cumulant generating function of Z can be expressed as*

$$K_Z(t) = \begin{cases} \log\left\{ \int_{-\infty}^{\infty} \exp(\mathrm{i}tz) \int_0^1 \frac{1}{w_1} f_{X_1}\left(\frac{z}{w_1}\right) f_{W_1}(w_1) dw_1 dz \right\}, & \text{if } n = 1, \\[2em] \log\left\{ \int_{T_n} \int_{-\infty}^{\infty} \exp(\mathrm{i}tz) \frac{\partial}{\partial z} \left[ \int_{B_z} c_X(u_1, \ldots, u_n) du_1 \cdots du_n \right] dz c_W(v_1, \ldots, v_{n-1}) dv_1 \cdots dv_{n-1} \right\}, \\ \quad \text{if } n \geq 2, \end{cases}$$

*where* $\mathrm{i} = \sqrt{-1}$ *and* $c_X$ *and* $c_W$ *are densities of copulas.*

**Proof.** Immediate from Theorem A14. $\quad\square$

**Theorem A16.** *With the stated assumptions in Section 5, the mth moment of Z can be expressed as*

$$E(Z^m) = \begin{cases} \int_{-\infty}^{\infty} z^m \int_0^1 \frac{1}{w_1} f_{X_1}\left(\frac{z}{w_1}\right) f_{W_1}(w_1) dw_1 dz, & \text{if } n = 1, \\[2em] \int_{T_n} \int_{-\infty}^{\infty} z^m \frac{\partial}{\partial z} \left[ \int_{B_z} c_X(u_1, \ldots, u_n) du_1 \cdots du_n \right] dz c_W(v_1, \ldots, v_{n-1}) dv_1 \cdots dv_{n-1}, \\ \quad \text{if } n \geq 2, \end{cases}$$

*where* $c_X$ *and* $c_W$ *are densities of copulas.*

**Proof.** The result follows from

$$E(Z^m) = \int_{-\infty}^{\infty} z^m f_Z(z) dz$$

and Theorem 11. $\quad\square$

**Theorem A17.** *With the stated assumptions in Section 5, the value at risk of Z with probability p can be expressed as the root of*

$$\int_0^1 F_{X_1}\left(\frac{\mathrm{VaR}_Z(p)}{w_1}\right) f_{W_1}(w_1) dw_1 = p \tag{A5}$$

*if $n = 1$ and as the root of*

$$\int_{T_n} \int_{B_{\text{VaR}_Z(p)}} c_X(u_1, \ldots, u_n) c_W(v_1, \ldots, v_{n-1}) du_1 \cdots du_n dv_1 \cdots dv_{n-1} = p \qquad (\text{A6})$$

*if $n \geq 2$, where $c_X$ and $c_W$ are densities of copulas.*

**Proof.** (A5) and (A6) follow immediately from Theorem 10. □

**Theorem A18.** *With the stated assumptions in Section 5, the expected shortfall of Z with probability $p$ can be expressed as*

$$\text{ES}_p(Z) = \begin{cases} \dfrac{1}{p} \displaystyle\int_0^1 \int_{-\infty}^{\frac{\text{VaR}_p(Z)}{w_1}} w_1 x_1 f_{X_1}(x_1) f_{W_1}(w_1) dx_1 dw_1, & \text{if } n = 1, \\[2em] \dfrac{1}{p} \displaystyle\int_{T_n} \int_{B_{\text{VaR}_Z(p)}} \left[ F_{W_1}^{-1}(v_1) F_{X_1}^{-1}(u_1) + \cdots + \left( 1 - \sum_{i=1}^{n-1} F_{W_i}^{-1}(v_i) \right) F_{X_n}^{-1}(u_n) \right] \\[1em] \qquad c_X(u_1, \ldots, u_n) c_W(v_1, \ldots, v_{n-1}) du_1 \cdots du_n dv_1 \cdots dv_{n-1}, \\ \qquad \text{if } n \geq 2, \end{cases} \qquad (\text{A7})$$

*where $c_X$ and $c_W$ are densities of copulas.*

**Proof.** (A7) follows immediately from Theorem 13. □

**Appendix D. Supplementary Results for Weighted Distribution IV**

**Theorem A19.** *With the stated assumptions in Section 6, the moment generating function of Z can be expressed as*

$$M_Z(t) = P(N = 1) \int_0^1 \frac{1}{w_1} \int_{-\infty}^{\infty} \exp(tz) f_{X_1}\left( \frac{z}{w_1} \right) dz f_{W_1}(w_1) dw_1$$

$$+ \sum_{n=2}^{\infty} P(N = n) \int_{T_n} \int_{-\infty}^{\infty} \exp(tz) \frac{\partial}{\partial z} \left\{ \int_{B_{z,n}} c_X(u_1, \ldots, u_n) du_1 \cdots du_n \right\} dz$$

$$c_W(v_1, \ldots, v_{n-1}) dv_1 \cdots dv_{n-1},$$

*where $c_X$ and $c_W$ are densities of copulas.*

**Proof.** The result follows from

$$M_Z(t) = E[\exp(tZ)] = \int_{-\infty}^{\infty} \exp(tz) f_Z(z) dz$$

*and Theorem 15.* □

**Theorem A20.** *With the stated assumptions in Section 6, the characteristic function of Z can be expressed as*

$$\phi_Z(t) = P(N = 1) \int_0^1 \frac{1}{w_1} \int_{-\infty}^{\infty} \exp(\text{i}tz) f_{X_1}\left( \frac{z}{w_1} \right) dz f_{W_1}(w_1) dw_1$$

$$+ \sum_{n=2}^{\infty} P(N = n) \int_{T_n} \int_{-\infty}^{\infty} \exp(\text{i}tz) \frac{\partial}{\partial z} \left\{ \int_{B_{z,n}} c_X(u_1, \ldots, u_n) du_1 \cdots du_n \right\} dz$$

$$c_W(v_1, \ldots, v_{n-1}) dv_1 \cdots dv_{n-1},$$

*where $c_X$ and $c_W$ are densities of copulas.*

**Proof.** The result follows from

$$\phi_Z(t) = E[\exp(\mathrm{i}tZ)] = \int_{-\infty}^{\infty} \exp(\mathrm{i}tz) f_Z(z) dz$$

and Theorem 15. □

**Theorem A21.** *With the stated assumptions in Section 6, the cumulant generating function of Z can be expressed as*

$$K_Z(t) = \log \left\{ P(N = 1) \int_0^1 \frac{1}{w_1} \int_{-\infty}^{\infty} \exp(\mathrm{i}tz) f_{X_1}\left(\frac{z}{w_1}\right) dz f_{W_1}(w_1) dw_1 \right.$$
$$+ \sum_{n=2}^{\infty} P(N = n) \int_{T_n} \int_{-\infty}^{\infty} \exp(\mathrm{i}tz) \frac{\partial}{\partial z} \left\{ \int_{B_{z,n}} c_X(u_1, \ldots, u_n) du_1 \cdots du_n \right\} dz$$
$$\left. c_W(v_1, \ldots, v_{n-1}) dv_1 \cdots dv_{n-1} \right\},$$

*where $c_X$ and $c_W$ are densities of copulas.*

**Proof.** Immediate from Theorem A20. □

**Theorem A22.** *With the stated assumptions in Section 6, the mth moment of Z can be expressed as*

$$E(Z^m) = P(N = 1) \int_0^1 \frac{1}{w_1} \int_{-\infty}^{\infty} z^m f_{X_1}\left(\frac{z}{w_1}\right) dz f_{W_1}(w_1) dw_1$$
$$+ \sum_{n=2}^{\infty} P(N = n) \int_{T_n} \int_{-\infty}^{\infty} z^m \frac{\partial}{\partial z} \left\{ \int_{B_{z,n}} c_X(u_1, \ldots, u_n) du_1 \cdots du_n \right\} dz$$
$$c_W(v_1, \ldots, v_{n-1}) dv_1 \cdots dv_{n-1},$$

*where $c_X$ and $c_W$ are densities of copulas.*

**Proof.** The result follows from

$$E(Z^m) = \int_{-\infty}^{\infty} z^m f_Z(z) dz$$

and Theorem 15. □

**Theorem A23.** *With the stated assumptions in Section 6, the value at risk of Z with probability p can be expressed as the root of*

$$P(N = 1) \int_0^1 F_{X_1}\left(\frac{\mathrm{VaR}_p(Z)}{w_1}\right) f_{W_1}(w_1) dw_1$$
$$+ \sum_{n=2}^{\infty} P(N = n) \int_{T_n} \int_{B_{\mathrm{VaR}_Z(p),n}} c_X(u_1, \ldots, u_n) c_W(v_1, \ldots, v_{n-1}) du_1 \cdots du_n dv_1 \cdots dv_{n-1} = p, \tag{A8}$$

*where $c_X$ and $c_W$ are densities of copulas.*

**Proof.** (A8) follows immediately from Theorem 14. □

**Theorem A24.** *With the stated assumptions in Section 6, the expected shortfall of Z with probability p can be expressed as*

$$
\begin{aligned}
\mathrm{ES}_p(Z) = {}& \frac{P(N=1)}{p} \int_0^1 \int_{-\infty}^{\frac{\mathrm{VaR}_p(Z)}{w_1}} w_1 x_1 f_{X_1}(x_1) f_{W_1}(w_1) dx_1 dw_1 \\
& + \frac{1}{p} \sum_{n=2}^{\infty} P(N=n) \int_{T_n} \int_{B_{\mathrm{VaR}_Z(p),n}} \left[ F_{W_1}^{-1}(v_1) F_{X_1}^{-1}(u_1) + \cdots + \left( 1 - \sum_{i=1}^{n-1} F_{W_i}^{-1}(v_i) \right) F_{X_n}^{-1}(u_n) \right] \\
& \qquad c_X(u_1, \ldots, u_n) c_W(v_1, \ldots, v_{n-1}) du_1 \cdots du_n dv_1 \cdots dv_{n-1},
\end{aligned}
\tag{A9}
$$

*where $c_X$ and $c_W$ are densities of copulas.*

**Proof.** (A9) follows immediately from Theorem 17. $\square$

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
