# Peer review of "Exact Results for the Distribution of Randomly Weighted Sums"

_mathematics, doi:10.3390/math12010149_

Round 1
Reviewer 1 Report
Comments and Suggestions for Authors
Dear authors,
In this paper, they have derived formulae for the cumulative distribution function, probability density function, conditional expectation, moment generating function, characteristic function, cumulant generating function, mth moment, value at risk and expected shortfall of Z, considering four possible situations. i) (X1, . . . ,Xn) are dependent and (W1, . . . ,Wn−1) are independent; ii) (X1, . . . ,XN) are dependent conditioned on N, (W1, . . . ,WN−1) are independent conditioned on N and N itself is a random variable; iii) (X1, . . . ,Xn) are dependent and (W1, . . . ,Wn−1) are dependent too; iv) (X1, . . . ,XN) are dependent conditioned on N, (W1, . . . ,WN−1) are dependent too conditioned on N and N itself is a random variable.
I consider it important to consider examples for each case.
Reviewer 2 Report
Comments and Suggestions for Authors
Please see the attached file.

Reviewer 3 Report
Comments and Suggestions for Authors
In this article, the authors study "Exact results for the distribution of randomly weighted sums".
The manuscript is very long, the authors should reduce the paper and leave the most significant results. Some developments can be sent as an appendix.
The following points are suggestions for improving the manuscript.
1) In the introduction the authors should better motivate the applicability of their results.
2) The references are quite recent and that is good enough, but also the authors should include some not-so-recent references with respect to weighted distribution.
3) The authors should incorporate some examples or applications where these results can be used.
4) Authors should write the article in the journal format.
Round 2
Reviewer 2 Report
Comments and Suggestions for Authors
The paper may be accepted in its present form.